# Enhancing Water Status and Nutrient Uptake in Drought-Stressed Lettuce Plants (*Lactuca sativa* L.) via Inoculation with Different *Bacillus* spp. Isolated from the Atacama Desert

**DOI:** 10.3390/plants13020158

**Published:** 2024-01-06

**Authors:** Christian Santander, Felipe González, Urley Pérez, Antonieta Ruiz, Ricardo Aroca, Cledir Santos, Pablo Cornejo, Gladys Vidal

**Affiliations:** 1Departamento de Ciencias Químicas y Recursos Naturales, Universidad de La Frontera, Temuco 4811230, Chile; c.santander01@ufromail.cl (C.S.); f.gonzalez31@ufromail.cl (F.G.); u.perez01@ufromail.cl (U.P.); maria.ruiz@ufrontera.cl (A.R.); cledir.santos@ufrontera.cl (C.S.); 2Grupo de Ingeniería Ambiental y Biotecnología, Facultad de Ciencias Ambientales y Centro EULA-Chile, Universidad de Concepción, Concepción 4070411, Chile; 3Programa de Doctorado en Ciencias Mención Biología Celular y Molecular Aplicada, Universidad de La Frontera, Temuco 4811230, Chile; 4Departamento de Microbiología del Suelo y la Planta, Estación Experimental del Zaidín, CSIC, Profesor Albareda 1, 18008 Granada, Spain; ricardo.aroca@eez.csic.es; 5Escuela de Agronomía, Facultad de Ciencias Agronómicas y de los Alimentos, Pontificia Universidad Católica de Valparaíso, Quillota 2260000, Chile; 6Centro Regional de Investigación e Innovación para la Sostenibilidad de la Agricultura y los Territorios Rurales, CERES, La Palma, Quillota 2260000, Chile

**Keywords:** drought, native *Bacillus* strains, plant growth promoting rhizobacteria (PGPR)

## Abstract

Drought is a major challenge for agriculture worldwide, being one of the main causes of losses in plant production. Various studies reported that some soil’s bacteria can improve plant tolerance to environmental stresses by the enhancement of water and nutrient uptake by plants. The Atacama Desert in Chile, the driest place on earth, harbors a largely unexplored microbial richness. This study aimed to evaluate the ability of various *Bacillus* sp. from the hyper arid Atacama Desert in the improvement in tolerance to drought stress in lettuce (*Lactuca sativa* L. var. capitata, cv. “Super Milanesa”) plants. Seven strains of *Bacillus* spp. were isolated from the rhizosphere of the Chilean endemic plants *Metharme lanata* and *Nolana jaffuelii*, and then identified using the 16s rRNA gene. Indole acetic acid (IAA) production, phosphate solubilization, nitrogen fixation, and 1-aminocyclopropane-1-carboxylic acid (ACC) deaminase activity were assessed. Lettuce plants were inoculated with *Bacillus* spp. strains and subjected to two different irrigation conditions (95% and 45% of field capacity) and their biomass, net photosynthesis, relative water content, photosynthetic pigments, nitrogen and phosphorus uptake, oxidative damage, proline production, and phenolic compounds were evaluated. The results indicated that plants inoculated with *B. atrophaeus*, *B. ginsengihumi*, and *B. tequilensis* demonstrated the highest growth under drought conditions compared to non-inoculated plants. Treatments increased biomass production and were strongly associated with enhanced N-uptake, water status, chlorophyll content, and photosynthetic activity. Our results show that specific *Bacillus* species from the Atacama Desert enhance drought stress tolerance in lettuce plants by promoting several beneficial plant traits that facilitate water absorption and nutrient uptake, which support the use of this unexplored and unexploited natural resource as potent bioinoculants to improve plant production under increasing drought conditions.

## 1. Introduction

The unprecedented rise in temperature has led to a surge in the occurrence of droughts, erratic precipitation patterns, heatwaves, and other extraordinary phenomena worldwide; these climatic conditions have a direct impact on agricultural production [1]. In this sense, world food security is one of the most important challenges of the 21st century; we aim to supply enough food to the growing population, but this is currently strongly threatened by global warming, especially the increase in drought periods [2]. Currently, drought is maybe the most major challenge for agriculture worldwide, as it is one of the main causes of global losses in vegetable and food products. It reduces the average yield of most crops and has significant economic impacts [3]. To meet the increasing demand for food by a growing global population, with an emphasis on producing a sufficient quantity and quality of food products, the current drought scenario will encourage a transformation of agricultural methods [4]. In plants, physiological drought is produced when water absorption is reduced by the roots, and it not replenish water losses in the leaves, causing tissue dehydration, which is crucial to avoid, when taking into account that between 80 and 95% of the fresh weight of plants consists of water [5].

To improve plant responses to drought stress, different approaches are currently being studied, such as traditional breeding methods, transgenic technology, and priming methods, among others. However, each method has some problems and limitations, because of the complexity of drought effects on plants and the responses of plants to drought stress [6,7]. An alternative that has generated vast interest in recent years is the use of plant growth-promoting rhizobacteria (PGPR) as inoculants (biofertilizers) in agriculture [8]. PGPR have the ability to directly enhance plant growth by promoting the absorption of nutrients from the soil, including processes such as nitrogen fixation, iron sequestration, and phosphate solubilization [9]. Additionally, PGPR produce various metabolites, such as IAA and hydrogen cyanide (HCN), which are synthesized by bacteria that further benefit the plants [10]. Various studies have reported that PGPR improve plant tolerance to abiotic stresses by producing a root–soil interface that enhances the absorption of water and nutrients [11]. Different bacteria genera, including Acetobacter, Azospirillum, Azotobacter, Bacillus, Burkholderia, Klebsiella, Pseudomonas, and Serratia have been recognized as PGPR [12]. Among the various species of PGPR, *Pseudomonas* and *Bacillus* spp. have been identified as being predominant in soil communities [13].

Due to their ability to thrive in diverse biotic and abiotic environments, *Bacillus* spp. have extensively been studied [14]. These bacteria have been found to enhance plant resilience to drought by facilitating increased water absorption, which serves as a crucial mechanism to protect plants against drought-induced damage [15]. Moreover, their root and soil colonization induce physiological alterations, such as elevated stomatal conductance, enhanced photosynthesis, increased levels of photosynthetic pigments, reduced lipid peroxidation, and beneficial nutritional changes, including the augmented uptake of nitrogen, phosphorus, and potassium. These combined effects contribute to mitigating the negative impacts of drought [16,17,18]. In addition, *Bacillus* spp. have the ability to synthesize plant growth-promoting substances such as IAA, gibberellins, and cytokinins, among others, which have been shown to stimulate cell division and elongation in both roots and shoots [19,20]. Moreover, they also secrete ACC deaminase, which plays a role in inhibiting ethylene synthesis in crop plants and promoting plant growth [21].

The native plant microbiome of extreme environments could hold a great potential as a source of stress-tolerant microorganisms, which could help to mitigate the impact of various environmental stressors on crop plants [22]. The microbial ecology of extreme environments, like deserts, is an intriguing area of research. The severe conditions in these environments have led to the evolution of microbial communities that are specially adapted to survive under such extreme circumstances [23]. In this context, the Atacama Desert in Chile is well-known as the driest place on Earth’s surface, which harbors largely unexplored microbial communities. Studies have demonstrated that certain bacterial groups in extreme environments can significantly influence the growth and flowering of plants [24]. Overall, while the study of the microbiome in extreme environments is still in its early stages, there is great potential for the development of novel microbiome engineering techniques that could help ensure global food security in an eco-friendly and sustainable way [25].

Based on the above, our hypothesis was raised that the inoculation of some efficient plant-growth promoting (PGP) *Bacillus* spp. isolated from extreme conditions ameliorates the negative effects of drought on plants cropped under water starvation conditions. To evaluate this hypothesis, comprehensive analyses were conducted to determine the impact of native drought-tolerant *Bacillus* isolates on the regulation of water and nutrient uptake, as well as the modulation of secondary metabolites associated with oxidative stress conditions in lettuce plants. The novelty of this study lies in an innovative approach to address drought stress in horticultural crops, specifically lettuce, based on the extensive consumption of this species, through the inoculation with *Bacillus* strains characterized as PGP microorganisms. These strains were isolated from ecosystems facing restricted water conditions, exemplified by the challenging environment of the Atacama Desert. This strategy emerges as a promising alternative to confront the mega-drought projected in the coming years for global agriculture. The findings from this study not only will contribute to an advanced comprehension of the mechanisms governing the response of plants associated with PGPR under drought conditions, but also lay the groundwork for the development of robust and efficient agricultural management strategies resilient to such stressors, based in the exploration and exploitation of drought-adapted microbial resources.

## 2. Results

### 2.1. Bacillus Identification

A total of seven *Bacillus* strains were isolated from the Atacama Desert. Among these, five isolates were obtained from the rhizosphere of the endemic plant *Metharme lanata,* specifically named ATMLC22021, ATMLC92021, ATMLC152021, ATMLC42021, and ATMLC102021. The remaining two strains, ATNJC12015 and ATNJC22015, were obtained during a desert bloom in 2015 from the rhizosphere of *Nolana jaffuelii*. Each strain was cultured from an isolated colony in liquid LB medium and then harvested for DNA extraction. Subsequently, the ribosomal 16S gene was amplified, sequenced, and subjected to a BLAST analysis in the GenBank database. The closest relative sequences and *Bacillus* strain sequences were used to construct a phylogenetic tree, using the neighbor-joining method, with the sequence of *Paenibacillus polymyxa* serving as the outgroup (Figure 1). The *Bacillus* strains were classified into six distinct clades. Among those isolated from the rhizosphere of *Metharme lanata*, strains ATMLC92021 and ATMLC152021 were classified within the *Bacillus subtilis* clade.

The sequences demonstrated a similarity of 99.3% in the BLAST analysis for ATMLC92021 and 100% for ATMLC152021 when compared to *Bacillus subtilis* APU-T03 (LC520136.1) and *Bacillus subtilis* MN180946.1 (MT111040.1), respectively. Strain ATMLC102021 was grouped in the *Bacillus tequilensis* clade and was closely related to *Bacillus tequilensis* FJAT-46257 (MK860003.1), with a similarity of 16S ribosomal sequence of 100%. Similarly, strains ATMLC42021 and ATMLC22021 were grouped in the *Bacillus frigoritolerans* and *Bacillus megaterium* clade, showing a similarity of 97% with *Bacillus frigoritolerans* FJAT-44608 (KX767092.1) and *Bacillus megaterium* CanS-59 (KT580611.1), respectively.

Regarding the strain isolated from the rhizosphere of *Nolana jaffuelii*, strain ATNJC22015 was identified as *Bacillus ginsengihumi* and shared a sequence similarity of 99% with *Bacillus ginsengihumi* strain BG6 (OQ947098.1). Meanwhile, strain ATNJC12015 was grouped in the *Bacillus atrophaeus* clade and was closely related to *Bacillus atrophaeus* LOCK 1013 (KT728844.1), with a similarity of 16S ribosomal gene sequence of 100%. All obtained sequences were submitted to the GenBank database with the following accession numbers: ATMLC22021 (OR243305), ATMLC42021 (OR243306), ATMLC92021 (OR243307), ATMLC152021 (OR243308), ATNJC22015 (OR243309), ATNJC12015 (OR243310), and ATMLC102021 (OR243311).

### 2.2. Plant Growth Promotion Properties 

The production of IAA dependent on tryptophan was assessed for seven strains of *Bacillus* spp., with the maximum reached by *Bacillus frigoritolerans* ATMLC42021 at 25.84 ± 2.30 µg mL^−1^, and the minimum shown by *Bacillus ginsengihumi* ATNJC22015 at 7.86 ± 0.76 µg mL^−1^ (Figure 2). Among the *Bacillus* species with lower production of tryptophan-dependent IAA, no differences were found between *B. atrophaeus* ATNJC12015, *B. ginsengihumi* ATNJC22015, and *B. subtilis* ATMLC152021. In contrast, in comparison to *B. megaterium* ATMLC22021 and *B. tequilensis* ATMLC102021, a statistically higher amount was determined for the former one.

Relevant traits for PGP traits in microorganisms include a nitrogen fixation capacity, phosphate solubilization, and the production of the enzyme ACC deaminase (see Appendix A). To assess these traits, the seven strains of *Bacillus* were evaluated under in vitro conditions (Table 1). As a result, the strains *B. atrophaeus* ATNJC12015, *B. ginsengihumi* ATNJC22015, *B. tequilensis* ATMLC102021, *B. subtilis* ATMLC92021, and *B. subtilis* ATMLC152021 exhibited positive results in all evaluated traits. However, *B. frigoritolerans* ATMLC42021 and *B. megaterium* ATMLC22021 were found to lack the capacity for phosphate solubilization and ACC deaminase production but were positive for nitrogen fixation.

### 2.3. Biomass Production and Relative Water Content

The fresh weight of shoots (FWS; Figure 3A) and relative water content (RWC; Figure 3B) were strongly influenced by the inoculation type (IN) and water regimen (WR) (*p* < 0.05). However, only the RWC was influenced by the interaction of the two factors (*p* < 0.05). The greater value of FWS for the well-watered (WW) condition was observed in plants inoculated with *B. tequilensis* ATMLC102021 (28.33 ± 1.41 g), and the lowest value was observed in the plants inoculated with *B. subtilis* ATMLC92021 (20.39 ± 0.25 g). Furthermore, under the WW regime, plants inoculated with *B. atrophaeus* ATNJC12015, *B. subtilis* ATMLC92021, and *B. subtilis* ATMLC152021 showed lower values than non-inoculated plants (control).

Under water stress conditions (WS), the highest values of FWS were observed for the plants inoculated with *B. tequilensis* ATMLC102021, *B. atrophaeus* ATNJC12015, and *B. ginsengihumi* ATNJC22015, which were larger than 20%, 31%, and 45%, respectively, compared to non-inoculated plants under drought stress (19.25 ± 0.45 g). The RWC under WW conditions presented the lowest value in the plants inoculated with *B. frigoritolerans* ATMLC42021 (81.95 ± 1.81%), and the highest value was observed in the plants inoculated with *B. ginsengihumi* ATNJC22015 (93.08 ± 1.81%). On the other hand, under WS conditions, the RWC reached the maximum value in the plants inoculated with *B. ginsengihumi* ATNJC22015 (91.29 ± 1.24%), being statistically higher than non-inoculated plants (81.32 ± 2.78%), which had the lowest value. The same effect was observed for plants inoculated with *B. atrophaeus* ATNJC12015, *B. ginsengihumi* ATNJC22015, *B. megaterium* ATMLC22021, and *B. subtilis* ATMLC92021.

### 2.4. Nitrogen and Phosphorus Uptake

The nitrogen (N, Figure 4A) and phosphorus (P, Figure 4B) contents in the leaves of lettuce plants were strongly influenced by IN and the interaction of both factors (INxWR) (*p* < 0.0001), while P uptake was also influenced by WR (*p* < 0.001). Under WW conditions, the N content in the leaves of lettuce plants had no statistical differences compared to non-inoculated plants, in the treatments with *B. ginsengihumi* ATNJC22015, *B. megaterium* ATMLC22021, *B. subtilis* ATMLC92021, and *B. subtilis* ATMLC152021, and lower values were observed in the plants inoculated with *B. atrophaeus* ATNJC12015, *B. frigoritolerans* ATMLC42021, and *B. tequilensis* ATMLC102021.

Under WS conditions, the N content was higher in the plants inoculated with different *Bacillus* species, by at least 21% more than non-inoculated plants (9.3 ± 0.44 mg g DW^−1^). The major N content in leaves of plants inoculated with *Bacillus* species was observed in the treatment with *B. atrophaeus* ATNJC12015 (13.2 ± 0.44 mg g DW^−1^), and the lowest values were observed in the plants inoculated with *B. tequilensis* ATMLC102021 (11.31 ± 0.28 mg g DW^−1^).

Under WW conditions, the P content was higher, in the range from 55% to 81%, for the plants inoculated with *B. atrophaeus* ATNJC12015, *B. ginsengihumi* ATNJC22015, *B. megaterium* ATMLC22021, *B. subtilis* ATMLC92021, and *B. subtilis* ATMLC152021, compared to non-inoculated plants (2.74 ± 1.01 mg g DW^−1^). In the case of WS conditions, the P content in the leaves of lettuce plants showed no statistical differences compared to non-inoculated plants in the treatments with *B. ginsengihumi* ATNJC22015 and *B. subtilis* ATMLC92021, while for the treatments with *B. atrophaeus* ATNJC12015, *B. megaterium* ATMLC22021, *B. subtilis* ATMLC152021, *B. frigoritolerans* ATMLC42021, and *B. tequilensis* ATMLC102021, the P content for the leaves of lettuce plants was higher by 190% to 323%, compared to non-inoculated plants.

### 2.5. Photosynthetic Behavior and Pigments

The content of chlorophyll A was not influenced by IN, WR, or their interaction (Figure 5A). In contrast, the chlorophyll B (Figure 5B), total chlorophyll (Figure 5C), and carotenoids (Figure 5D) were influenced by IN (*p* < 0.05), and total chlorophyll was also influenced by InxWR (*p* < 0.001). In the case of chlorophyll B, under WW, the plants inoculated with *B. subtilis* ATMLC152021 showed greater content, compared to non-inoculated plants, while under WS conditions, the content of chlorophyll B was improved in the leaves by the inoculations with *B. ginsengihumi* ATNJC22015 and *B. atrophaeus* ATNJC12015 by 140% and 134%, respectively, compared to non-inoculated plants.

Under WW conditions, total chlorophyll was improved, in comparison to non-inoculated plants, by the inoculation with *B. subtilis* ATMLC152021. Under WS conditions, total chlorophyll was improved by 2 and 1.84 times more for the treatments with *B. ginsengihumi* ATNJC22015 and *B. atrophaeus* ATNJC12015, respectively, than non-inoculated plants. In the case of carotenoids, no significant differences were found between WS and WW conditions with the inoculation of different *Bacillus* species.

The net photosynthesis (A; Figure 6A) was strongly influenced by IN (*p* < 0.0001). Stomatal conductance (gs; Figure 6B) was strongly influenced by IN, WR, and the interaction of both factors (InxWR; *p* < 0.0001). Under WW, the A was higher in the plants inoculated with *B. atrophaeus* ATNJC12015 and *B. ginsengihumi* ATNJC22015, compared to the non-inoculated plants. Under WS, the net photosynthesis had the highest value in the plants inoculated with *B. ginsengihumi* ATNJC22015, being 3.2 times higher than non-inoculated plants. For the treatments with *B. subtilis* ATMLC92021, *B. subtilis* ATMLC152021, *B. tequilensis* ATMLC102021, and *B. atrophaeus* ATNJC12015, there was observed at least two times the value obtained in non-inoculated plants under WS. Also, no differences were observed when comparing plants inoculated with *B. atrophaeus* ATNJC12015, *B. megaterium* ATMLC22021, *B. subtilis* ATMLC92021, *B. subtilis* ATMLC152021, *B. frigoritolerans* ATMLC42021, and *B. tequilensis* ATMLC102021 between WW and WS regimens. 

Higher values of A were observed in plants under WS compared to WW in the treatment with *B. ginsengihumi* ATNJC22015. The gs, under WW conditions, showed a lower value only in the plants inoculated with *B. tequilensis* ATMLC102021, with non-statistical differences observed in the remaining treatments, in comparison with non-inoculated plants. Under WS conditions, the plants inoculated with different *Bacillus* species had higher values in comparison to non-inoculated plants, with the highest value observed in the treatments with *B. ginsengihumi* ATNJC22015, *B. megaterium* ATMLC22021, *B. subtilis* ATMLC92021, and *B. tequilensis* ATMLC102021, with at least two times higher values than control plants. Besides, in these treatments, no differences were observed in the gs values under WS and WW conditions.

### 2.6. Oxidative Damage and Proline Production

The proline content (Figure 7A) and malondialdehyde content (Figure 7B; TBARS) were influenced by the IN source of variation (*p* < 0.001). In the case of proline content, it was also influenced by the interaction of both factors (INxWR; *p* < 0.001). Meanwhile, TBARS were also influenced by WR (*p* < 0.001). Under WW regime, malondialdehyde content showed no statistical differences between control plants and the treatments with *B. subtilis* ATMLC92021, *B. frigoritolerans* ATMLC42021, *B. tequilensis* ATMLC102021, and *B. atrophaeus* ATNJC12015.

In the treatments with *B. atrophaeus* ATNJC12015, *B. ginsengihumi* ATNJC22015, *B. megaterium* ATMLC22021, and *B. subtilis* ATMLC152021, no differences were observed from the WS condition, and a lower value of malondialdehyde content was obtained, compared to non-inoculated plants. The proline content was higher in all treatments in WS conditions than in WW conditions. Under the WW condition, no differences were found between plants inoculated with different *Bacillus* species and non-inoculated plants. However, under the WS regime, the plants inoculated with *B. atrophaeus* ATNJC12015, *B. megaterium* ATMLC22021, *B. subtilis* ATMLC152021, and *B. frigoritolerans* showed a decrease in proline content compared to non-inoculated plants. Besides, in the treatments with *B. ginsengihumi* ATNJC22015, *B. subtilis* ATMLC92021 showed no statistical differences compared to non-inoculated plants under the WS condition.

### 2.7. Identification, Quantification, and Antioxidant Capacity of Phenolic Compounds

Through High-performance liquid chromatography with diode array detection and electrospray ionization tandem mass spectrometry (HPLC-DAD-ESI-MS/MS), six phenolic compounds were identified (Table 2), and seven phenolic compounds were quantified by HPLC–DAD. Their concentrations were determined by comparing them with standards of chlorogenic acid and quercetin-3-glucoside. This was measured through the absorbance at wavelengths of 320 nm and 360 nm, respectively (Figure 8). Among these, 5-caffeoylquinic acid was the compound with the greatest concentration in leaves, ranging from 501.67 ± 169.75 to 296.26 ± 81.70 µg g FW^−1^, while the lowest concentration of phenolic compounds was observed for Quercetin-hexoside, ranging from 17.68 ± 2.52 to 7.01 ± 0.91 µg g FW^−1^. Among these phenolic compounds, Quercetin-hexoside (Peak 4) and non-identified compounds (Peak 2) with a retention time of 9.00 were influenced by the level of inoculation (*p* < 0.05). Caffeoylquinic acid (peak 1) and dicaffeoylquinic acid (peak 7) were not influenced by any factor. For Quercetin-hexoside, no differences were observed in the control plants between the WW and WS conditions. However, under the WW condition, plants inoculated with *B. atrophaeus* ATNJC12015, *B. megaterium* ATMLC22021, and *B. tequilensis* ATMLC102021 (B.t) demonstrated a significantly lower concentration in lettuce leaves. Similarly, a decrease in the concentration of Quercetin-hexoside was observed in plants subjected to WS and inoculated with *B. megaterium* ATMLC22021. Non-identified phenolic compounds were less expressed in lettuce plants by the inoculation of *B. atrophaeus* ATNJC12015, *B. megaterium* ATMLC22021, and *B. tequilensis* ATMLC102021 under WW conditions, compared to control plants. Similarly, a decrease in concentration was observed under the WS condition by the inoculation of *B. megaterium* ATMLC22021, compared to control plants under WS.

The antioxidant activity (AA), which was determined by the 2,2-diphenyl-1-picrylhydrazyl (DPPH; Figure 9A) method, the Trolox equivalent antioxidant capacity method (TEAC; Figure 9B), and by the copper reducing antioxidant capacity method (CUPRAC; Figure 9C), and total phenols, which were determined by the Folin–Ciocalteu method (TPC; Figure 9D), were influenced by the IN and the interaction of both factors (INxWR; *p* < 0.001). Under WW conditions, the AA determined by DPPH and TEAC showed lower values in the treatment with *B. tequilensis* ATMLC102021 and *B. megaterium* ATMLC22021, compared to non-inoculated plants. In the case of AA determined by CUPRAC, a higher value was observed in the plants inoculated by *B. subtilis* ATMLC92021 and *B. subtilis* ATMLC152021, and no statistically significant differences were observed in the remaining treatments compared to non-inoculated plants under WW conditions.

On the other hand, under WS conditions, the AA determined by DPPH decreased compared to control plants in the treatments with *B. atrophaeus* ATNJC12015, *B. ginsengihumi* ATNJC22015, and *B. megaterium* ATMLC22021. In the case of AA determined by TEAC, a lower value was observed in plants inoculated with *B. atrophaeus* ATNJC12015, *B. ginsengihumi* ATNJC22015, and *B. subtilis* ATMLC92021, compared to non-inoculated plants. For AA determined by CUPRAC under the WS regimen, a decrease was observed in the plants inoculated with *B. atrophaeus* ATNJC12015, *B. ginsengihumi* ATNJC22015, *B. megaterium* ATMLC22021, *B. subtilis* ATMLC92021, and *B. frigoritolerans* ATMLC42021, compared to control plants under the WS regimen.

The total phenolic compounds showed no statistical differences when comparing control plants under the WS and WW regimens. A similar trend was observed in the treatments with *B. atrophaeus* ATNJC12015, *B. ginsengihumi* ATNJC22015, *B. megaterium* ATMLC22021, and *B. subtilis* ATMLC92021. Under the WW regimen, a decrease was observed in the total phenolic compounds only for the treatment with *B. tequilensis* ATMLC102021. Meanwhile, under the WS regimen, the total phenolic compounds decreased in the plants inoculated with *B. atrophaeus* ATNJC12015, *B. ginsengihumi* ATNJC22015, *B. megaterium* ATMLC22021, and *B. subtilis* ATMLC92021.

### 2.8. Multivariate Analysis

Multivariate analysis was performed separately, categorized by water regimen (WR). In the case of the WW condition, the two first principal components (PCs) explained 54.50% of the observed cumulative variance, with 38.07% and 16.43% in the first (PC1) and second (PC2) principal components, respectively (Figure 10A). In PC1, the variables that contributed most positively were the total phenolic compounds, the phenolic compounds Peak3, Peak4, Peak5, and Peak7, and the AA determined by the TEAC and DPPH. Such variables were associated with the treatment of plants inoculated with *B. ginsengihumi* ATNJC22015, *B. subtilis* ATMLC92021, and *B. frigoritolerans* ATMLC42021.

On the other hand, the main variable that contributed negatively in PC1 under WW conditions was proline content, which was associated with the plants inoculated with *B. megaterium* ATMLC22021 and *B. atrophaeus* ATNJC12015. PC2 under WW conditions was most positively influenced by the variables chlorophyll A, chlorophyll B, total chlorophyll, and P uptake, and was associated with the treatment with *B. subtilis* ATMLC152021. Additionally, the variables that most negatively influenced PC2 under WW conditions were FWS and carotenoids content, and these were mainly associated with the inoculation of *B. tequilensis* ATMLC102021.

Under WS conditions, PC1 and PC2 explained 38.13% and 16.43% of the cumulative variance, respectively (Figure 10B). In PC1, the variables with the highest weight that contributed positively were the total phenolic compounds (TPC), DPPH, CUPRAC, TEAC, and the seven identified phenolic compounds, whereas the variables that contributed negatively were the total chlorophyll content, chlorophyll A and B, and P and N uptake. In PC2, the variables that contributed most positively were chlorophyll A, peak 4, peak 5, total chlorophyll content, and fresh shoot weight.

In contrast, the variables carotenoids, proline, and TBARS were the ones that contributed the most negatively. In this sense, the plants inoculated with *B. ginsengihumi* ATNJC22015 and *B. atrophaeus* ATNJC12015, which showed the highest fresh shoot weight production under drought stress conditions, were strongly associated with the total chlorophyll content, chlorophyll A, chlorophyll B, RWC, A, gs, and N uptake. On the other hand, the control plants, which exhibited lower biomass production under drought stress, were associated with CUPRAC, TEAC, and MDA content (TBARS).

## 3. Discussion

In this study, we isolated and identified seven strains of the genus *Bacillus* through ribosomal 16S gene sequencing. Among them, the strains ATMLC92021 and ATMLC152021 were closely related to *Bacillus subtilis*. In previous studies, some strains of this species have been reported to improve the drought tolerance in sugarcane, fenugreek, and wheat [26,27,28]. In addition, the strain ATMLC102021 was closely related to *B. tequilensis* clade; this species was described as osmotic stress-tolerant and presented with the capacity to improve drought stress tolerance in maize, wheat, and eggplant [29,30]. The strain ATMLC22021 was grouped within the clade of *B. megaterium*. This species has extensively been studied [31,32,33,34,35]. In genome analysis, it was reported to possess multiple genes that contribute to plant growth under drought conditions, such as genes related to IAA synthesis, P-solubilization, siderophore and volatile organic compound production, and the synthesis of osmoprotectants such as trehalose, glycine/betaine, and spermidine. These traits were the key factors in maintaining the growth of the host plant under abiotic stress, especially drought [31]. Reinforcing this, previous works have reported an improvement in drought tolerance due to the presence of *B. megaterium* in mulberry, wheat, cabbage, maize and rice [31,32,33,34,35]. Moreover, the strains ATNJC12015, ATNJC22015, and ATMLC42021 were grouped in *B. atrophaeus, B. ginsengihumi*, and *B. frigoritolerans,* respectively. For these species, there were no previous studies reporting an improvement in drought tolerance for plants. However, for *B. atrophaeus* and *B. frigoritolerans*, salt stress tolerance in maize was reported [36,37]. Similarly, various traits were reported for *B. ginsengihumi*, such as IAA production ability, N fixation ability, P-solubilization capacity, and siderophore production ability [38,39]. In this sense, the *Bacillus* species isolated herein show great potential to enhance drought tolerance in plants.

Drought stress negatively impacts on both plant growth and productivity by affecting various physiological and biochemical processes, including photosynthesis, chlorophyll synthesis, nutrient metabolism, respiration, and carbohydrate metabolism. All these alterations are translated into a significant reduction in biomass production [40]. In the present study, water deficit had a negative impact on plant performance, resulting in a decreased plant biomass production in both inoculated and non-inoculated plants. Plants employ a variety of strategies to overcome drought stress, including a combination of stress avoidance and regulation of drought tolerance, depending on their genotype [41]. In the same way, research has shown the role of root-associated microbial communities in enhancing plant drought tolerance. This emphasizes the role of interaction between plants and the soil microbiomes inhabiting the rhizosphere and root system, which is recognized as a pivotal factor in plants’ rapid adaptation to abiotic stress [42].

Several PGPR have been documented to trigger drought tolerance in plants by causing various physiological and biochemical changes [43]. *Bacillus* species, such as *B. megaterium*, *B. circulans*, *B. coagulans*, *B. subtilis*, *B. azotofixans*, *B. macerans*, *B. velezensis*, and others, are reported as PGPR that improve plant growth under abiotic stress [44]. In agreement with the above findings, the results of our study showed that the inoculation with *B. ginsengihumi* ATNJC22015, *B. atrophaeus* ATNJC12015, and *B. tequilensis* ATMLC102021 has a significant positive effect on lettuce growth, promoting a higher biomass accumulation of plants. The use of *Bacillus* as a rhizobacterium inoculant that promotes plant growth has extensively been recorded across a wide range of horticultural crops [45]. For instance, *B. subtilis* Rhizo SF 48 enhanced tomato plant growth traits caused by drought-stress [21]. In the same way, the total dry weight of lettuce plants was significantly improved by inoculation with *Bacillus* sp. and *B. subtilis*, compared to the controls, under 50% and 33% field capacity conditions [46].

Plants’ capacity to survive and thrive in times of drought stress can be attributed to their adaptive mechanisms, which involve enhanced nutrient absorption and transportation [47]. It is a widely recognized fact that drought stress can lead to nutrient deficiencies by significantly affecting the plant’s ability to uptake nutrients, even in soils that have been fertilized [48]. Due to various reasons, this can result in nutrient deficiencies, such as a reduction in nutrient supply through mineralization and decreased mobility and absorbance of individual nutrients. Therefore, the rate of mineral diffusion from the soil matrix to the roots is lower [49]. It is well known that N and P are primary macronutrients that play critical roles in various processes of plant growth [50], and previous studies have demonstrated that drought stress greatly decreases their concentrations [51,52]. Under drought conditions, PGPR have been demonstrated as having functions as biofertilizers, directly promoting plant growth by improving nutrient acquisition, through processes like N fixation, mineral solubilization, and P and potassium (K) absorption [53]. In the present study, the capacity to fix the atmospheric N_2_ and P-solubilization activity was tested in Nfb and Pikovskaya’s agar, respectively. Furthermore, our results demonstrated that all strains assessed had the capacity to fix atmospheric N_2_, which aligns with previously reported findings on *Bacillus* species [54,55,56].

The ability to convert atmospheric N_2_ into ammonia by PGPR is an essential trait for selecting bacteria as bioinoculants to enhance drought stress tolerance [57]. Our findings showed that, under drought conditions, all *Bacillus* species led to increased N concentrations in lettuce plants, compared to non-inoculated plants. Similar findings were observed in sugarcane when inoculated with *B. subtilis* [58], as well as in maize, following inoculation with *Bacillus* spp., under drought stress conditions [59]. Nitrogen, an essential nutrient for crop growth and development, also plays a critical role in regulating hydraulic conductance and supporting photosynthesis during drought conditions [60,61]. The increased N uptake has been associated with enhanced crop drought tolerance, emphasizing the complex connection between N metabolism and water absorption [62]. In the same way, P is a key player in the growth and development of plants. Nevertheless, the restricted accessibility of P in soil poses substantial hurdles for crop productivity, particularly under the influence of abiotic stressors like drought, salinity, and extreme temperatures [63].

When PGPR is employed to enhance P-solubility, it facilitates P uptake by the roots, ultimately enhancing the plant’s resilience to abiotic stress [64]. In the present work, the strains *B. atrophaeus* ATNJC12015, *B. ginsengihumi* ATNJC22015, *B. tequilensis* ATMLC102021, *B. subtilis* ATMLC92021, and *B. subtilis* ATMLC152021 have the capacity to solubilize phosphates on Pikovskaya’s agar medium. Similarly, several *Bacillus* species, including *B. circulans*, *B. cereus*, *B. fusiformis*, *B. pumilus*, *B. megaterium*, *B. mycoides*, *B. coagulans*, *B. chitinolyticus*, and *B. subtilis*, have been documented as P-solubilizers [65]. Plants inoculated with *B. atrophaeus* ATNJC12015, *B. tequilensis* ATMLC102021, and *B. subtilis* ATMLC152021 showed an interesting behavior in P uptake. Plants exhibited an in vitro P-solubilization capacity and increased P levels in shoots under drought stress. In contrast, *B. megaterium* ATMLC22021 and *B. frigoritolerans* ATMLC42021 did not show a P-solubilization capacity; however, the plants inoculated with these strains also displayed increased P concentrations in shoots under drought conditions, compared to non-inoculated plants. Overall, we hypothesize that both *Bacillus* strains enhance P concentration in lettuce by upregulating root P-transporters, rather than through increased P availability via solubilization. This could be supported by previous reports by Romer-Munar and Aroca [66], who found that the *B. megaterium* strain was unable to solubilize K^+^, but it modified the expression of K^+^ transporters under K^+^-deficit stress conditions, thereby ameliorating the nutritional status of rice plants.

Aside from their capacity to convert nutrients into a usable form, soil bacteria also release phytohormones, such as auxins, cytokinins, gibberellins, and enzymes, as ACC-deaminase, which are responsible for stimulating plant growth [67]. In this work, all *Bacillus* species isolated produced IAA in the presence of tryptophan, within a range from 25.84 ± 2.30 µg mL^−1^ to 7.86 ± 0.76 µg mL^−1^ (Figure 1). These results are in agreement with the high capacity of *Bacillus* species to produce IAA in an LB medium, with a mean of less than 30 µg mL^−1^ [68]. Indole-3-Acetic Acid (IAA), a natural form of auxin, is a crucial plant growth regulator involved in various growth processes. IAA plays a role in cell division, elongation, differentiation, gene expression, and photosynthesis initiation, contributing to the development of lateral and adventitious roots [69]. Furthermore, rhizobacteria aid in enhancing plant drought stress resilience by providing IAA, and they can also regulate IAA production in plants. When plant roots release L-tryptophan, a major precursor for IAA, PGPR in the rhizosphere convert tryptophan into IAA, which is then taken up by the plants [70]. PGPR can independently produce their own IAA, which is absorbed by plant cells, activating the IAA signal transduction pathway, promoting plant cell growth [57]. As mentioned above, the presence of IAA released by PGPR is an important trait that helps plants to cope with various abiotic stresses, such as drought. However, a higher content of IAA can increase the biosynthesis of ethylene by stimulating ACC synthetase, which is involved in leaf abscission and senescence under drought stress, thus reducing crop yields [37,71]. To overcome this, PGPR can produce 1-aminocyclopropane-1-carboxylate deaminase (ACC deaminase), an enzyme that facilitates the breakdown of the cyclopropane ring and the removal of the amino group from ACC, the direct precursor of ethylene. Consequently, this process leads to a reduction in the level of ethylene in plants and an improvement in plant drought resistance [72,73].

The presence of ACC activity in *Bacillus* species is an extensive subject of study, mainly associated with resistance to various abiotic stress factors, such as salinity, high temperatures, the presence of heavy metals, and drought [74,75,76,77]. Particularly, the presence of *Bacillus* ACC activity was correlated with an improvement in drought stress tolerance in different crops such as fenugreek, tomato, wheat, and corn [21,78,79,80]. In the present study, the strains *B. atrophaeus* ATNJC12015, *B. ginsengihumi* ATNJC22015, *B. tequilensis* ATMLC102021, *B. subtilis* ATMLC92021, and *B. subtilis* ATMLC152021 were found to produce ACC deaminase (Table 1). Interestingly, the strains *B. frigoritolerans* ATMLC42021 and *B. megaterium* ATMLC22021, which exhibited a higher capacity for IAA production, did not exhibit ACC deaminase activity. Furthermore, plants inoculated with these strains did not demonstrate an improvement in drought tolerance, revealing that PGPR traits were essential to overcome water scarcity.

Leaf RWC is a crucial indicator of plant water deficit stress tolerance. Under drought stress, the RWC decreases, with the extent of reduction increasing as stress severity escalates [47]. Numerous studies have indicated that beneficial soil microorganisms enhance plant resilience to abiotic stresses by establishing a root–soil interface that facilitates water uptake [11,81]. For example, bacterial inoculation has demonstrated that it improves water use efficiency, root and shoot biomass, RWC, and membrane stability index, effectively mitigating the adverse effects of drought stress in wheat and tomato plants [82]. Similar results were observed here, where drought stress led to a significant reduction in leaf RWC in plants. However, the inoculation of lettuce plants with *B. ginsengihumi* ATNJC22015 and *B. atrophaeus* ATNJC12015 resulted in increased RWC and higher levels of photosynthetic pigments, indicating an effective frontline defense against drought stress.

According to Gowtham et al. [21], assessing the levels of RWC and photosynthetic pigments in plants undergoing bacterial treatment is a key method for evaluating the positive impacts of treatments on plants experiencing drought stress. Furthermore, the higher chlorophyll content and RWC in plants may be linked to the well-preserved chloroplast structure [21]. It is well-known that, under drought stress, the closure of stomata reduces gs, which reduces water loss, but it ultimately reduces photosynthesis [83]. In the current study, all *Bacillus* species increased gs, compared to non-inoculated plants, under drought conditions. Likewise, it is remarkable that *B. ginsengihumi* ATNJC22015 produced a higher increase in A in lettuce plants, improving both under well-irrigated and drought conditions compared to non-inoculated plants. Consequently, PGPR delay stomatal closure and transpiration under water stress by enhancing water absorption and they exert a positive influence on the activity of the photosynthetic apparatus, resulting in a smaller decline in photosynthesis [84].

Due to impaired electron transport in mitochondria, the plasma membrane, and chloroplasts, plants produce reactive oxygen species (ROS) when subjected to drought stress conditions. Hydrogen peroxide (H_2_O_2_) is generated as a result of photorespiration, and these ROS induce oxidative damage in plants [85]. In our study, the values determined as oxidative damage (malondialdehyde concentration; TBARS) were the lowest in comparison with other studies. Notably, our study revealed a reduction in MDA production in plants inoculated with *B. atrophaeus* ATNJC12015, *B. ginsengihumi* ATNJC22015, *B. megaterium* ATMLC22021, and *B. subtilis* ATMLC152021, compared to non-inoculated plants. This effect was observed in both WW and WS conditions. Plants respond to oxidative stress by accumulating compatible organic solutes like proline, glycine betaine, and phenolic compounds, among others [86,87]. Proline accumulates under stress and acts as a crucial cellular redox buffer [88]. Proline helps to detoxify ROS, serves as a regulatory osmotic agent, preserves membrane integrity, and stabilizes antioxidant enzymes [89]. However, in the present study, inoculation with any *Bacillus* spp. significantly diminished the accumulation of this amino acid under drought conditions, compared to non-inoculated plants.

Phenolic compounds, a significant category of plant secondary metabolites, may undergo an increase in their contents under stress conditions, thereby playing a role in safeguarding against ROS, regulating osmotic balance, or fortifying the overall defense mechanisms of plants [90]. The synthesis of phenolic compounds in lettuce is contingent upon the genotype and stress intensity, and can be influenced by the presence of PGPR microorganisms [91,92]. In this way, our results showed that both drought stress and *Bacillus* inoculation did not significantly affect the production of phenolic compounds or their antioxidant activity. In summary, plants have evolved to cope with both biotic and abiotic stresses by employing specialized metabolites. Nevertheless, the precise role of these metabolites in stress responses remains a subject of debate, contingent upon factors such as the host plant, the nature of the stressor, and the resulting product.

## 4. Materials and Methods

### 4.1. Isolation of Bacillus *spp.*

The *Bacillus* species were isolated from the rhizosphere of two endemic plant of Chile belonging to the Zygophyllaceae (*Metharme lanata* Phill.) and Solanaceae (*Nolana jaffuelii* Johnst.) families from different sites in the Atacama Desert (20°24′34.628″ S, 82°7′51.727″ W Palo Buque; 20°51′17.813″ S, 81°10′10.927″ W Quebrada de Cahuiza; 21°6′31.727″ S, 81°6′31.493″ W; 21°6′31.59″ S, 81°6′32.26″ W; 21°6′36.681″ S, 81°8’44.973″ W Quebrada de Choja, Tarapacá Region, northern Chile). The soils were sampled from a 5–20 cm soil layer, collected aseptically using a metallic shovel disinfected with 70% ethanol, and stored in sterilized plastic bags. Samples were kept at 4 °C in a cooler and immediately transported to the laboratory for processing. A soil suspension was prepared by adding 1 g of soil to 49 mL of sterile NaCl 0.9% solution, followed by a 10-fold serial dilution. *Bacillus* spp. were isolated by inoculating 100 µL of each dilution on LB agar medium (tryptone 10 g L^−1^, yeast extract 5 g L^−1^, sodium chloride 10 g L^−1^) and then incubating the plates at 30 °C for 48 h. After purification by repeated streaking on the LB agar medium, different colonies were selected based on their morphological characteristics such as shape, color, and size. The isolated colonies were stored at −80 °C in a 50% glycerol solution for subsequent analysis. 

### 4.2. Molecular Identification of Bacillus *spp.*

Total genomic DNA was extracted from 2 mL of 2-day-old samples of each *Bacillus* strain cultured on LB broth medium, using a Wizard^®^ Genomic DNA Purification Kit (Promega, Madison, WI, USA) according to the manufacturer’s instructions. Polymerase chain reaction (PCR) was performed for the amplification of the 16S ribosomal DNA (rDNA) gene, using GoTaq^®^ DNA Polymerase (Promega, Madison, WI, USA) in a total volume of 25 μL per reaction with a final concentration of 1X colorless buffer, 3 mM MgCl_2_, 1 mM dNTPs, 0.4 μM of each universal primer for 16S ribosomal DNA sequence amplification (27F: 5′-AGAGTTTGATCCTGGCTCAG-3′ and 1492R: 5′-GGTTACCTTGTTACGACTT-3′) [93,94], 1.25 U per reaction of GoTaq^®^ DNA Polymerase, and approximately 0.25 μg of genomic DNA. The amplification conditions were as follows: 94 °C for 5 min for initial DNA denaturation, 35 cycles at 94 °C for 30 s, 60 °C for 30 s, and 72 °C for 30 s, and a final elongation step at 72 °C for 5 min. The amplified products were analyzed by gel electrophoresis. The PCR products were sequenced from both directions using the 16S rDNA Forward and Reverse primers on an automated DNA sequencer ABI PRISM 3500xL (Applied Biosystems, Palo Alto, CA, USA) by the sequencing service of Pontificia Universidad Católica, Santiago, Chile (CONICYT-FONDEQUIP EQM150077).

The sequences were assembled and manually edited in BioEdit v.7.2.5. The resulting nucleotide sequences were analyzed using the BLAST program [95] and compared to the NCBI GenBank database. The closely related *Bacillus* sequences were aligned with the obtained sequences using CLUSTALW (version 2.0.10) and then manually adjusted for phylogenetic analysis. The phylogenetic tree was constructed using the neighbor-joining method [96], by applying the Kimura-2-Parameter model [97] in the MEGA-X software, version 10.2.3 [98]. The confidence value of the node was supported by bootstrap analyses based on 1000 replications [99]. The sequences of the *Bacillus* spp. samples were submitted to the GenBank database.

### 4.3. Characterization of Plant Growth Promotion Properties

To characterize the PGP traits of each *Bacillus* strain, the ACC deaminase activity, IAA production, P-solubilization, and N fixation were determined. They were cultured in liquid LB media and incubated at 30 °C and 120 rpm for 48 h. After incubation, the culture was adjusted to 0.8 of OD600. The *Bacillus* isolates were tested for ACC deaminase activity using sterile minimal Dworkin and Foster (DF) salts media, supplemented with 3 mM ACC (1-cyclopropane-1-carboxylic acid) as the only N source. To begin with, each isolate (50 μL) was streaked on minimal DF salt agar plates. Additionally, a 50 μL suspension was streaked on ACC-deficient minimal DF salt agar plates as a negative control.

The plates were then incubated at 28 °C for 5 days. Colonies observed on the plates were identified as ACC deaminase producers [100]. The production of IAA was analyzed as described by Rangseekaew et al. [101]. Each *Bacillus* strain was cultured in LB and incubated at 30 °C and 120 rpm for 48 h. After incubation, the OD600 was set to 0.8. LB supplemented with 2 mg mL^−1^ of L-tryptophan (Sigma-Aldrich, Beijing, China) were inoculated with 100 µL of the adjusted *Bacillus* culture and incubated at 30 °C and 120 rpm in the dark.

The supernatant was collected by centrifugation at 11,000 rpm for 15 min and evaluated for IAA production using the colorimetric assay, by mixing 1 mL of the supernatant with 2 mL of Salkowski’s reagent [102]. After 30 min of incubation at room temperature in the dark, IAA production was detected spectrophotometrically by measuring the absorbance at 530 nm. The quantity of IAA production was determined from a standard curve using <99.9% pure IAA (Sigma-Aldrich, Beijing, China).

Screening for P-solubilizing activity was determined by inoculating 10 mL of the adjusted *Bacillus* culture in the center of Pikovskaya’s agar, with 0.5% (*w*/*v*) tri-calcium phosphate as the sole phosphate source [103], and incubating it for 7 days at 30 °C in the dark. A clear zone of tri-calcium phosphate solubilization around the bacterial colony indicates positive phosphate solubilization. N fixation was tested by inoculating 10 µL of the adjusted bacterial culture in NfB semisolid medium [104] and incubating for 7 days at 30 °C in the dark. Positive N fixation was observed in the presence of a turbid ring on the subsurface of the NfB medium. Three replicates were made for each strain in each determination.

### 4.4. Inoculum Preparation 

Each *Bacillus* strain was cultured in LB media broth and incubated at 30 °C and 120 rpm for 48 h. After incubation, the culture was centrifuged at 5000× *g* rpm for 5 min, and the supernatant was discarded. The bacterial pellet was resuspended in 50 mL of 10 mM MgSO_4_ and the OD_600_ was set to 0.8.

### 4.5. Experimental Design, Soil, and Biological Material

A fully randomized factorial 8 × 2 design was used. The first experimental factor was inoculation (IN), with the levels being: non-inoculated plants (control), plants inoculated with *B. atrophaeus* ATNJC12015 (B.a), *B. ginsengihumi* ATNJC22015 (B.g), *B. megaterium* ATMLC22021 (B.m), *B. subtilis* A ATMLC92021 (B.s_A), *B. subtilis* B ATMLC152021 (B.s_B), *B. frigoritolerans* ATMLC42021 (B.f), and *B. tequilensis* ATMLC102021 (B.t). For each level of inoculation, we used two watering regimes: (i) well-watered conditions (90–100% of available water capacity, WW), and (ii) plants subjected to water stress (40–50% of available water capacity, WS). A total of sixteen treatments with five replicates (n = 5) were performed (n = 80).

The growing substrate consisted of a mixture of peat moss and perlite (*v*/*v* 70:30%), and the mix was autoclave-sterilized at 121 °C for 60 min on 3 consecutive days. Lettuce seeds (*Lactuca sativa* L. var. capitata, cv. “Super Milanesa”) were surface sterilized using 5% NaClO for 5 min and then washed twice with sterile distilled water to eliminate NaClO residues. The sterilized seeds were sown in polystyrene trays, and each cell of the tray was inoculated directly with 1 mL of *Bacillus* inoculum. After 20 days, the seedlings were transplanted to 1 L pots containing 300 g of the peat moss and perlite mixture described above, and a second *Bacillus* inoculum was applied at that time. Non-inoculated plants received 1 mL of sterile 5 Mm MgSO_4_ solution in both cases.

### 4.6. Growth Conditions

The experiment was carried out for 60 days under greenhouse conditions (25/21 °C; 50/60% relative humidity, 14/10 h day/night photoperiod) at the greenhouses of the Universidad de La Frontera, Temuco, Chile. The watering regimes were applied 5 days post-transplantation via irrigation with the respective amount of tap water. Soil water content in the substrate of pots was measured in situ using GS3 sensors and EM 50 data loggers (Decagon^®^ Devices, Pullman, WA, USA).

### 4.7. Measurements in Plants

#### 4.7.1. Biomass Production, Nitrogen, and Phosphorus Uptake

At harvest, the shoot and root organs were weighed, and subsamples (3 g) of fresh material were ground in liquid N_2_ to obtain a fine powder that was stored at −80 °C for subsequent analysis. The residual material was dried (65 °C, 48 h) in a forced-air oven for chemical analysis. Subsequently, 1 g of ground shoot and root dry material was pulverized, incinerated at 550 °C, and finally digested in an acid mixture of H_2_O/HCl/HNO_3_ (8/1/1, *v*/*v*/*v*). The digests were used for the spectrophotometric determination of P using the blue-molybdate method. The total N content was determined by using the Kjeldahl digestion method, based in the distillation of NH_3_ and quantification by titration [105]. 

#### 4.7.2. Photosynthetic Parameters and Photosynthetic Pigments

The Targas-1 equipment (PP Systems, Amesbury, MA, USA) was used to determine photosynthetic traits, including the leaf internal concentration of CO_2_ (Ci: μmol mol^−1^), photosynthesis rate (A: μmol CO_2_ m^−2^ s^−1^), and stomatal conductance (gs: mmol H_2_O m^−2^ s^−1^), following the instructions provided in the user manual. Measurements were conducted on the second youngest leaf from four plants per treatment, one day before harvest. To extract photosynthetic pigments, 0.5 g of fresh material was mixed with 5 mL of methanol for 24 h. The resulting samples were then filtered using 13 mm diameter Millex filters with a 0.22 μm pore size nylon membrane (Millipore, Bedford, MA, USA). The absorbance of the filtered liquid was measured at wavelengths of 663 nm and 645 nm, corresponding to chlorophyll A and B, respectively, using a Synergy H1 Hybrid Multi-Mode microplate spectrophotometer (BioTek Inc., Winooski, VT, USA). The concentrations of pigments were calculated using the provided formula by Lichtenthaler [106].

#### 4.7.3. Oxidative Damage and Proline Production

Oxidative damage, which causes the peroxidation of lipids, is determined as an index of oxidative stress by measuring the concentration of thiobarbituric acid reactive substances (TBARS) in fresh leaves. To extract lipid peroxides, 100 mg of freshly powdered leaf tissue was mixed with 1.5 mL of 0.2% trichloroacetic acid (TCA) in 2 mL microtubes. The mixture was vortexed for 10 s and then centrifuged at 17,000× *g* for 10 min at 4 °C. To create the chromogen, 300 μL of the supernatant was combined with 1.2 mL of a mixture containing 20% (*w*/*v*) TCA and 0.5% 2-thiobarbituric acid. The final mixture was incubated at 95 °C for 30 min and then rapidly cooled in an ice bath. The resulting supernatants were subjected to spectrophotometric readings at 440 nm, 532 nm, and 600 nm, using a Synergy H1 Hybrid Multi-Mode microplate spectrophotometer (BioTek Inc., Winooski, VT, USA). The content of malondialdehyde (MDA) was calculated using the method described by Du and Bramlage [107]). The free proline concentration was determined using 0.05 g of fresh tissue and spectrophotometrically assayed at 530 nm, as described by Bates et al. [108].

#### 4.7.4. Identification and Quantification of Phenolic Compounds and Antioxidant Capacity

The extraction of phenolic compounds was conducted following the method described by Aguilera et al. [109], with minor modifications. Briefly, 0.3 g of leaf samples were placed in 5 mL of extraction solvent composed of methanol and formic acid (95:5 v:v). The samples were then subjected to sonication using an ultrasonic processor (Sonics and Materials, Newtown, CT, USA) at 130 W for 60 s with an amplitude of 40%. Subsequently, the samples were shaken for 30 min at 200 rpm and centrifuged at 4000× *g* for 10 min.

The resulting supernatant was transferred to a separate tube, protected from light, and stored at −20 °C until the antioxidant measurements were conducted. For HPLC determinations, the extracted solution was filtered using 0.22 µm pore filters and injected into amber vials. High-performance liquid chromatography–diode array detection (HPLC–DAD) analyses were performed using an HPLC system (Shimadzu, Tokyo, Japan) equipped with a quaternary LC-20AT pump (Shimadzu), a DGU-20A5R degassing unit (Shimadzu), a CTO-20A oven (Shimadzu), a SIL-20a autosampler (Shimadzu), and a UV-visible diode array spectrophotometer (SPD-M20A; Shimadzu). Instrument control and data collection were conducted using Lab Solutions software (version 5.96) (Shimadzu, Duisburg, Germany). The HPLC–DAD separation method for the determination of phenolic compounds, particularly hydroxycinnamic acids and flavonols, was developed based on the protocol described by Santander et al. [92]. Identification of the compounds was accomplished by comparing their MS/MS spectra with data from the literature and commercial standards. The MS/MS conditions employed were those reported by González et al. [110].

The Folin–Ciocalteu method, originally described by Singleton and Rossi [111], was employed to determine the concentration of total phenols. The method was adapted for use with a microplate reader [112]. The absorbance was measured at 750 nm, with gallic acid serving as the standard. The results were expressed as milligrams of gallic acid per gram of fresh weight (FW). To assess antioxidant activities, the DPPH (2,2-diphenyl-1-picrylhydrazyl) activity, TEAC (Trolox equivalent antioxidant activity), and CUPRAC (cupric ion reducing antioxidant activity) methods, as reported by Fritz et al. [113], were used.

### 4.8. Statistical Analysis

All statistical analysis and figures were performed in R version 4.2.1. An analysis of variance (ANOVA) was used to test for significant differences between measurements of each assay. For the variables with significant differences, the means were compared using the Fisher LSD multiple range test with the package “agricolae”, version 1.3.5. The dataset was split by water regimen (WW or WS) and subjected to a principal component analysis (PCA). Confidence ellipses (group means) by inoculation treatment were also generated using the packages “FactoMineR”, version 2.7, and “factoextra”, version 1.0.7.

## 5. Conclusions

Our results showed that the seven indigenous *Bacillus* species strains, isolated from the rhizosphere of two endemic plants growing in the hyper-arid Atacama Desert (Chile), exhibited plant growth-promoting activities. These activities included nitrogen fixation, phosphate solubilization, ACC deaminase activity, and indole-3-acetic acid (IAA) production. Likewise, the tolerance to drought in lettuce plants was improved by different *Bacillus* strains, with the best results achieved using *B. ginsengihumi* and *B. atrophaeus*. These strains demonstrated the potential to enhance water stress tolerance by improving nitrogen and water uptake. This enhancement resulted in improved photosynthetic performance, increased chlorophyll content, and higher relative water content levels, which were related to elevated biomass production. Our findings suggest that *Bacillus* strains isolated from the Atacama Desert hold promise as bioinoculants for enhancing crop production under drought conditions. Delving into the mechanisms underlying plant growth promotion by these bacteria and exploring their potential applications in agriculture is now a further crucial step.

## Figures and Tables

**Figure 1 plants-13-00158-f001:**
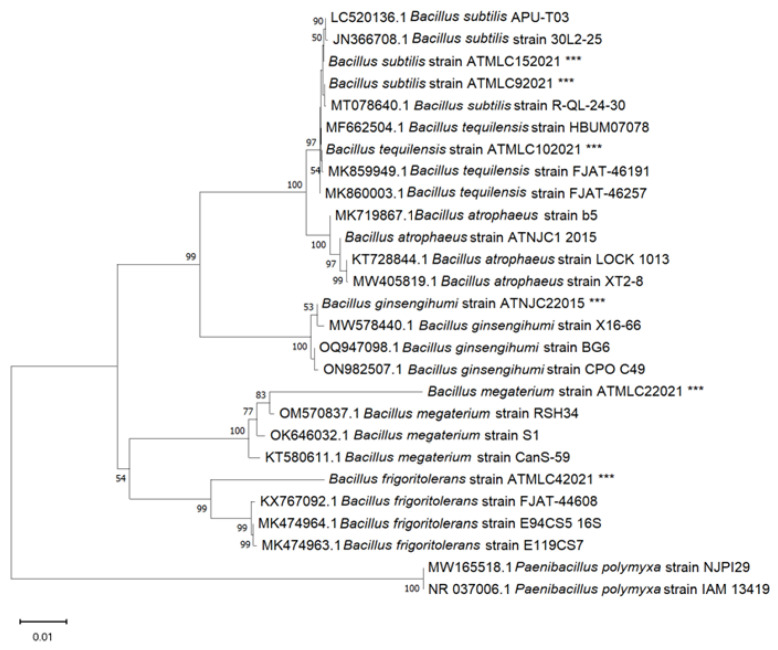
Phylogenetic tree obtained through the neighbor-joining method based on the 16S rRNA gene sequence. The confidence value of each node was supported by bootstrap analyses based on 1000 replications and is represented at each node. Bars represent 0.01 substitutions per nucleotide position. The species *Paenibacillus polymyxa* was used as the outgroup. The sequences obtained in this study were depicted with ***.

**Figure 2 plants-13-00158-f002:**
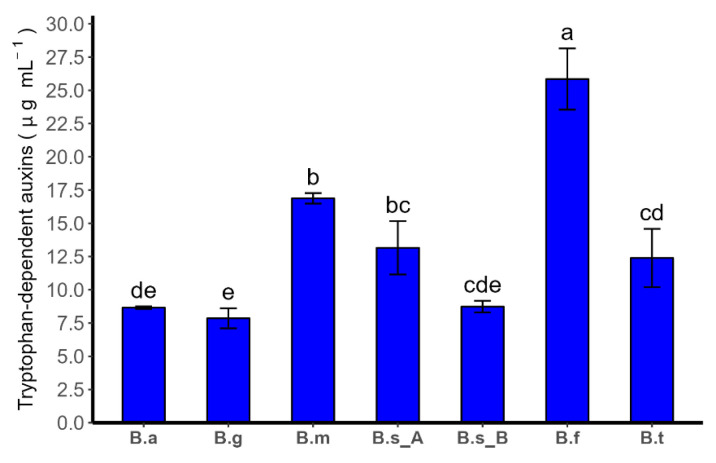
Tryptophan-dependent IAA production of the bacterial strains *Bacillus atrophaeus* ATNJC12015 (B.a), *B. ginsengihumi* ATNJC22015 (B.g), *B. megaterium* ATMLC22021 (B.m), *B. subtilis* ATMLC92021 (B.s_A), *B. subtilis* ATMLC152021 (B.s_B), *B. frigoritolerans* ATMLC42021 (B.f), and *B. tequilensis* ATMLC102021 (B.t). These strains were grown in LB media for 7 days at 30 °C and agitation to 120 rpm. The data obtained are presented as means ± standard error (SE), with a sample size of five. Statistical analysis was performed using a one-way ANOVA. Fisher’s LSD test was conducted to determine significant differences (*p* ≤ 0.05), which are denoted by different letters.

**Figure 3 plants-13-00158-f003:**
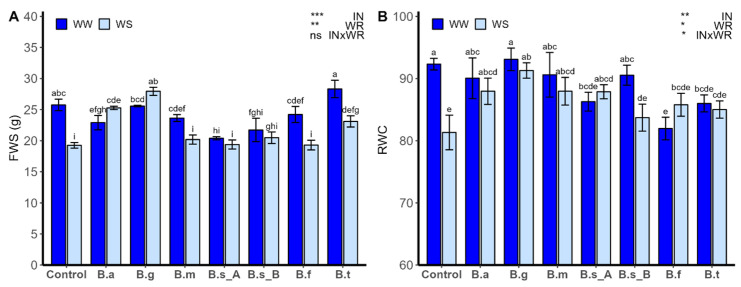
Fresh weight of shoots (FWS; (**A**)) and relative water content (RWC; (**B**)) of lettuce plants (*Lactuca sativa* L. var. capitata, cv. “Super Milanesa”) that are non-inoculated (control) or inoculated by *B. atrophaeus* ATNJC12015 (B.a), *B. ginsengihumi* ATNJC22015 (B.g), *B. megaterium* ATMLC22021 (B.m), *B. subtilis* ATMLC92021 (B.s_A), *B. subtilis* ATMLC152021 (B.s_B), *B. frigoritolerans* ATMLC42021 (B.f), or *B. tequilensis* ATMLC102021 (B.t) under non-stress condition (WW, dark blue) and under drought stress (WS, light blue) conditions. The data include means ± SE (n = 5) and were analyzed through a two-way ANOVA, conducted with inoculation (IN) and water regimen (WR) as sources of variation. The significant difference was depicted as ns: not significant, *p* < 0.05: *, *p* < 0.001: **, and *p* < 0.0001: ***. Different letters indicate significant differences (*p* ≤ 0.05), according to Fisher’s multiple range test.

**Figure 4 plants-13-00158-f004:**
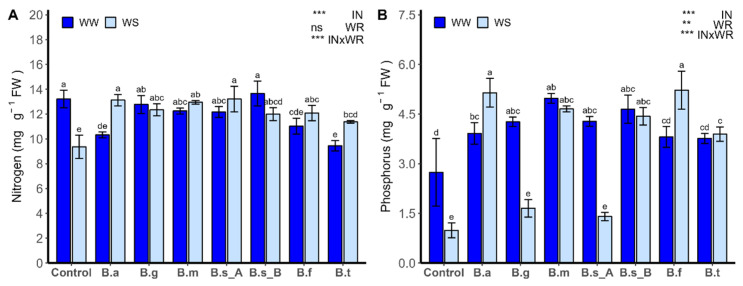
Nitrogen (**A**) and phosphorus uptake (**B**) in leaves of lettuce plants (*Lactuca sativa* L. var. capitata, cv. “Super Milanesa”) that are non-inoculated (control) or inoculated by *B. atrophaeus* ATNJC12015 (B.a), *B. ginsengihumi* ATNJC22015 (B.g), *B. megaterium* ATMLC22021 (B.m), *B. subtilis* ATMLC92021 (B.s_A), *B. subtilis* ATMLC152021 (B.s_B), *B. frigoritolerans* ATMLC42021 (B.f), or *B. tequilensis* ATMLC102021 (B.t) under non-stress condition (WW, dark blue) and drought stress (WS, light blue). The data include means ± SE (n = 5) and were analyzed through a two-way ANOVA, conducted with inoculation (IN) and water regimen (WR) as sources of variation. The significant difference was depicted as ns: not significant, *p* < 0.001: **, and *p* < 0.0001: ***. Different letters indicate significant differences (*p* ≤ 0.05), according to Fisher’s multiple range test.

**Figure 5 plants-13-00158-f005:**
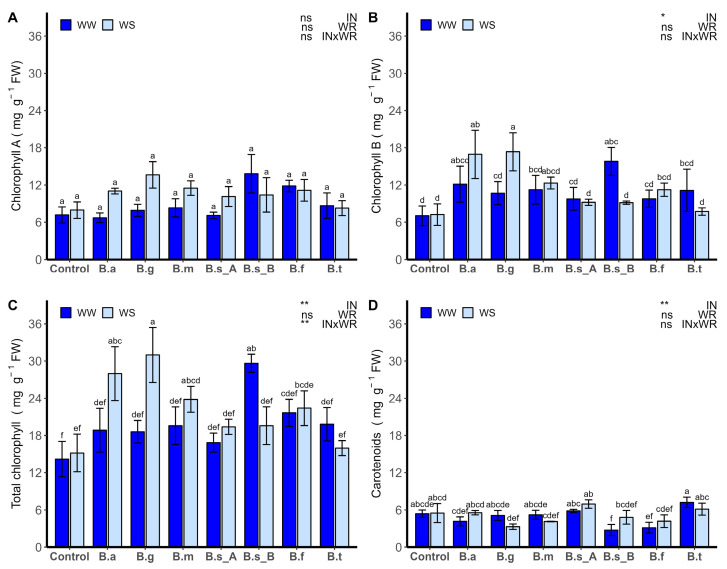
Chlorophyll A (**A**), chlorophyll B (**B**), total chlorophyll (**C**), and carotenoids (**D**) of lettuce plants (*Lactuca sativa* L. var. capitata, cv. “Super Milanesa”) that are non-inoculated (control) or inoculated by *B. atrophaeus* ATNJC12015 (B.a), *B. ginsengihumi* ATNJC22015 (B.g), *B. megaterium* ATMLC22021 (B.m), *B. subtilis* ATMLC92021 (B.s_A), *B. subtilis* ATMLC152021 (B.s_B), *B. frigoritolerans* ATMLC42021 (B.f), or *B. tequilensis* ATMLC102021 (B.t) under non-stress condition (WW, dark blue) and drought stress (WS, light blue). The data include means ± SE (n = 5) and were analyzed through a two-way ANOVA, conducted with inoculation (IN) and water regimen (WR) as sources of variation. The significant difference was depicted as ns: not significant, *p* < 0.01: *, and *p* < 0.001: **. Different letters indicate significant differences (*p* ≤ 0.05) according to Fisher’s multiple range test.

**Figure 6 plants-13-00158-f006:**
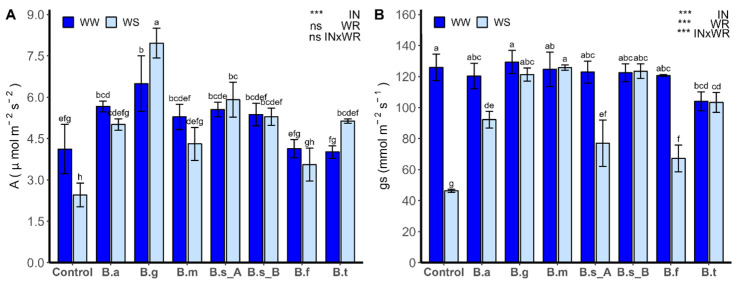
Net photosynthesis (**A**) and stomatal conductance (gs; (**B**)) of lettuce plants (*Lactuca sativa* L. var. *capitata*, cv. “Super Milanesa”) that are non-inoculated (control) or inoculated by *B. atrophaeus* ATNJC12015 (B.a), *B. ginsengihumi* ATNJC22015 (B.g), *B. megaterium* ATMLC22021 (B.m), *B. subtilis* ATMLC92021 (B.s_A), *B. subtilis* ATMLC152021 (B.s_B), *B. frigoritolerans* ATMLC42021 (B.f), or *B. tequilensis* ATMLC102021 (B.t) under non-stress condition (WW, dark blue), and under drought stress (WS, light blue). The data include means ± SE (n = 5) and were analyzed through a two-way ANOVA, conducted with inoculation (IN) and water regimen (WR) as sources of variation. The significant difference was depicted as ns: not significant, and *p* < 0.0001: ***. Different letters indicate significant differences (*p* ≤ 0.05) according to Fisher’s multiple range test.

**Figure 7 plants-13-00158-f007:**
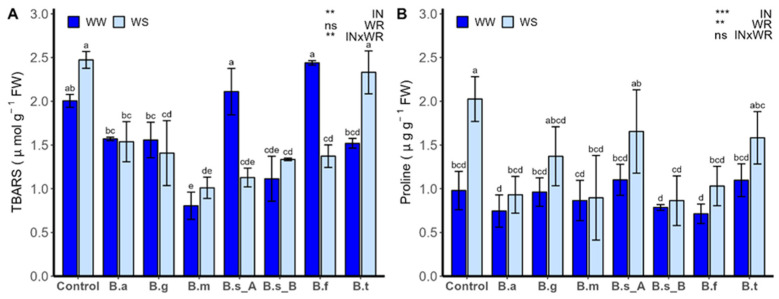
Proline content (**A**) and TBARS (**B**) in the leaves of lettuce plants that are non-inoculated (control) or inoculated by *B. atrophaeus* ATNJC12015 (B.a), *B. ginsengihumi* ATNJC22015 (B.g), *B. megaterium* ATMLC22021 (B.m), *B. subtilis* ATMLC92021 (B.s_A), *B. subtilis* ATMLC152021 (B.s_B), *B. frigoritolerans* ATMLC42021 (B.f), or *B. tequilensis* ATMLC102021 (B.t), under non-stress condition (WW, dark blue) and drought stress (WS, light blue). The data include means ± SE (n = 4) and were analyzed through a two-way ANOVA, conducted with inoculation (IN) and water regimen (WR) as sources of variation. The significant difference was depicted as ns: not significant, *p* < 0.001: **, and *p* < 0.0001: ***. Different letters indicate significant differences (*p* ≤ 0.05) according to Fisher’s multiple range test.

**Figure 8 plants-13-00158-f008:**
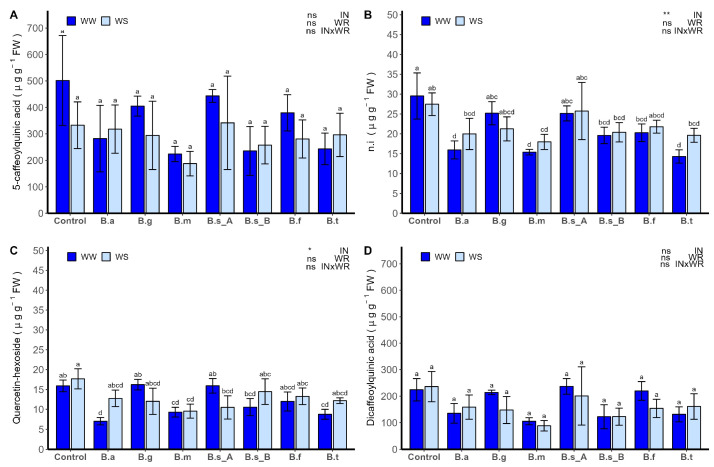
Principal phenolic compounds content, (**A**) 5-caffeoylquinic acid, (**B**) non-identified compounds (n.i), (**C**) Quercetin-hexoside, and (**D**) Dicaffeoylquinic acid in the leaves of lettuce plants (*Lactuca sativa* L. var. *capitata*, cv. “Super Milanesa”) that are non-inoculated (control) or inoculated by *B. atrophaeus* ATNJC12015 (B.a), *B. ginsengihumi* ATNJC22015 (B.g), *B. megaterium* ATMLC22021 (B.m), *B. subtilis* ATMLC92021 (B.s_A), *B. subtilis* ATMLC152021 (B.s_B), *B. frigoritolerans* ATMLC42021 (B.f), or *B. tequilensis* ATMLC102021 (B.t), under non-stress conditions (WW, dark blue) and drought stress conditions (WS, light blue). The data include means ± SE (n = 5) and were analyzed through a two-way ANOVA, conducted with inoculation (IN) and water regimen (WR) as sources of variation. The significant difference was depicted as ns: not significant, *p* < 0.01: *, and *p* < 0.001: **. Different letters indicate significant differences (*p* ≤ 0.05) according to Fisher’s multiple range test.

**Figure 9 plants-13-00158-f009:**
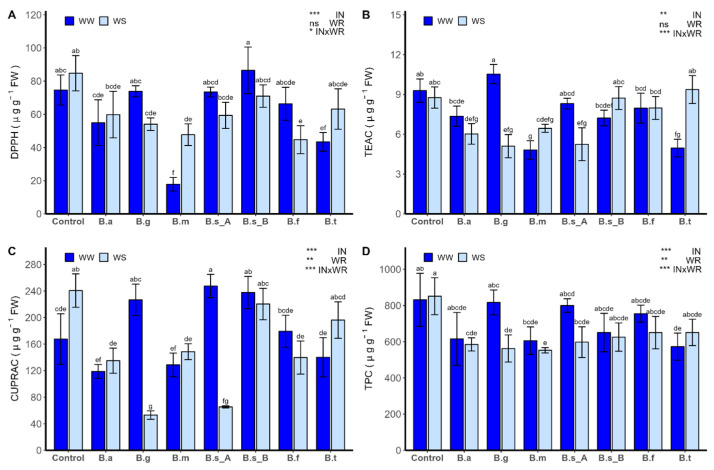
Phenolic compounds and antioxidant activities of leaves of lettuce plants (*Lactuca sativa* L. var. capitata, cv. “Super Milanesa”) that are non-inoculated (control) or inoculated by *B. atrophaeus* ATNJC12015 (B.a), *B. ginsengihumi* ATNJC22015 (B.g), *B. megaterium* ATMLC22021 (B.m), *B. subtilis* ATMLC92021 (B.s_A), *B. subtilis* ATMLC152021 (B.s_B), *B. frigoritolerans* ATMLC42021 (B.f), or *B. tequilensis* ATMLC102021 (B.t), under non-stress conditions (WW, dark blue), and drought stress conditions (WS, light blue). (**A**) Antioxidant activity (AA) determined by the DPPH (2,2-diphenyl-1-picrylhydrazyl) method, (**B**) AA determined by the TEAC (Trolox equivalent antioxidant capacity) method, (**C**) AA determined by the CUPRAC (copper reducing antioxidant capacity) method, and (**D**) total phenols determined by the Folin–Ciocalteu method (TPC). The data include means ± SE (n = 5) and were analyzed through a two-way ANOVA, conducted with inoculation and salinity stress as sources of variation. The significant difference was depicted as ns: non-significant, *p* < 0.05: *, *p* < 0.001: ** and *p* < 0.0001: ***. Different letters indicate significant differences (*p* ≤ 0.05) according to Fisher’s multiple range test.

**Figure 10 plants-13-00158-f010:**
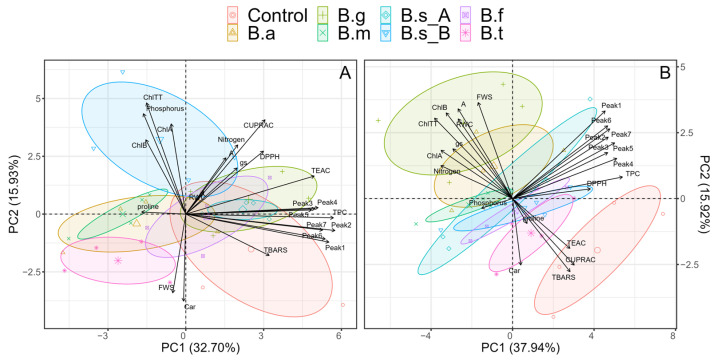
Principal components analysis (PCA) biplot for the water regimes of well-watered (WW; (**A**)) and with stress (WS; (**B**)) conditions, based on biomass production of shoots (FWS) and relative water content (RWC); photosynthetic parameters: net photosynthesis (**A**) and stomatal conductance (gs); photosynthetic pigments: chlorophyll A (ChlA), chlorophyll B (ChlB), total chlorophyll (ChlTT), and carotenoids (Car); total phenolic compounds (TPC); antioxidant activities (AA) determined by DPPH, TEAC, and CUPRAC methods; oxidative damage determined by TBARS; proline content (proline); nitrogen and phosphorus uptake in the leaf; and phenolic compounds determined by HPLC–DAD and MS/MS (Peak1, Peak2, Peak3, Peak4, Peak5, Peak6, Peak7, and Peak8). The confidence interval was depicted by different treatments of lettuce plants (*Lactuca sativa* L. var. capitata, cv. “Super Milanesa”); either non-inoculated (control) or inoculated by *B. atrophaeus* ATNJC12015 (B.a), *B. ginsengihumi* ATNJC22015 (B.g), *B. megaterium* ATMLC22021 (B.m), *B. subtilis* ATMLC92021 (B.s_A), *B. subtilis* ATMLC152021 (B.s_B), *B. frigoritolerans* ATMLC42021 (B.f), or *B. tequilensis* ATMLC102021 (B.t).

**Table 1 plants-13-00158-t001:** Phosphate (P) solubilization, nitrogen (N) fixation, and ACC deaminase activity of *Bacillus* strain isolated from the Atacama Desert.

*Bacillus* Strains	Nitrogen Fixation	Phosphate Solubilization	ACC Deaminase Production
*B. atrophaeus* ATNJC12015	+	+	+
*B. ginsengihumi* ATNJC22015	+	+	+
*B. frigoritolerans* ATMLC42021	+	−	−
*B. tequilensis* ATMLC102021	+	+	+
*B. megaterium* ATMLC22021	+	−	−
*B. subtilis* ATMLC92021	+	+	+
*B. subtilis* ATMLC152021	+	+	+

“+”: Positive result; “−”: Negative result.

**Table 2 plants-13-00158-t002:** Identification of phenolic compounds by HPLC-DAD-ESI-MS/MS in the leaves of lettuce plants (*Lactuca sativa* L. var. *capitata*, cv. “Super Milanesa”).

Peak	RT (min)	Compound	λ Max (nm)	[M − H]^−^	Product Ions
1	5.1	5-caffeoylquinic acid	326	353.1	191.1
2	9.0	n.i	-	431.2	295.0; 163.1
3	10.6	Coumaroylquinic acid	-	337.1	191.0
4	13.2	Quercetin-hexoside	351	463.1	300.0
5	14.3	Quercetin-acetylhexoside	355	505.1	300.0
6	15.3	Chicoric acid	329	473.1	311.0; 149.0
7	16.0	Dicaffeoylquinic acid	328	515.1	353.1; 191.0

## Data Availability

Data are contained within the article and Appendix A.

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
