# Peer review of "Enhancing Water Status and Nutrient Uptake in Drought-Stressed Lettuce Plants (Lactuca sativa L.) via Inoculation with Different Bacillus spp. Isolated from the Atacama Desert"

_plants, 2024, doi:10.3390/plants13020158_

Round 1

Reviewer 1 Report

Comments and Suggestions for Authors

The manuscript "Improving water status and nutrient uptake in lettuce plants under water stress by inoculation with different Bacillus spp. strains isolated from the Atacama Desert" demonstrates how inoculation with certain strains of Bacillus spp. can significantly improve the ability of lettuce plants to absorb water and nutrients under drought conditions. The research is of great importance for agriculture and global economy. It can be accepted after a minor revision.

1.      How were the Bacillus spp. strains selected for this study?

2.      How could this research be applied in large-scale agriculture?

3.      The needless “-” in the caption of Figure 1 should be deleted.

Author Response

The manuscript "Improving water status and nutrient uptake in lettuce plants under water stress by inoculation with different Bacillus spp. strains isolated from the Atacama Desert" demonstrates how inoculation with certain strains of Bacillus spp. can significantly improve the ability of lettuce plants to absorb water and nutrients under drought conditions. The research is of great importance for agriculture and global economy. It can be accepted after a minor revision.

R: Thank you for your positive comment.

  1. How were the Bacillus spp. strains selected for this study?

R: These were all the Bacillus species that we isolated from various rhizosphere soil samples taken from two plant species in the Atacama Desert. While actin, pseudomonads, among other genera, were also isolated, in this study, we chose to focus on Bacillus species. Bacillus spp. were selected based on morphology and subsequently identified through 16S sequencing.

  1. How could this research be applied in large-scale agriculture?

R: The identification and isolation of Bacillus species, as described in this study, may have promising large-scale applications in agriculture. Some of these isolated species demonstrated the ability to enhance drought tolerance by serving as plant growth-promoting agents. This could be part of broader soil management strategies and sustainable agricultural practices, contributing to global food security and production efficiency. Implementing the knowledge of Bacillus species on a large scale in agriculture involves mass production of beneficial strains, formulation of practical products, development of application protocols, extensive field trials, continuous monitoring of impacts, and sustainable practices. Education, technology transfer, collaboration, and compliance with regulations are key to success in this initiative. Nevertheless, initial research into the beneficial properties of Bacillus species suggests considerable potential to enhance sustainability and productivity in agriculture.

  1. The needless “-” in the caption of Figure 1 should be deleted.

R: Done, please see figure 1.

Reviewer 2 Report

Comments and Suggestions for Authors

Comments and Suggestions from my side.
The authors declaimed that Our results 36 show that specific Bacillus species from the Atacama Desert enhance drought stress tolerance in lettuce plantsby promoting several beneficial plant traits that facilitate water absorption, and nutrient uptake, which support its use as potent bioinoculants.

The authors need to revise the English grammar and spelling of the article. There are many incomprehensible phrases. I do not understand the novelty of the article.

Title. What do you mean? You must specify the lettuce scientific name you studied. "

Abstract Line 30. Write the scientific name of the lettuce  variety/cultivars you studied.

Pay attention to the use of points, commas, spaces, and capital letters in the whole manuscript.

Before the use of PGPR, you must explain it in the text

Introduction. Explain the problem in more detail. Provide references and explain what makes your study different.

Materials and methods. In each method explain with a phrase the reason why you performed this test.

Results: Phosphorus solubilization test should be suppourte by fotographs of halozone formation. Please share it as regular or supplementary figure.

In IAA production, After 30 minutes of incubation at 708 room temperature in the dark, IAA production was detected spectrophotometrically by 709 measuring the absorbance at 530 nm. But acutally here Pink colour is the indicator. Please share the pink colour in test tubes as supplementary figure

Conclusions must be clear and must show the novelty of your article.

Comments on the Quality of English Language

 Dear Editor

Comments and Suggestions from my side.
The authors declaimed that Our results 36 show that specific Bacillus species from the Atacama Desert enhance drought stress tolerance in lettuce plants by promoting several beneficial plant traits that facilitate water absorption, and nutrient uptake, which support its use as potent bioinoculants.

The authors need to revise the English grammar and spelling of the article. There are many incomprehensible phrases. I do not understand the novelty of the article.

Title. What do you mean? You must specify the lettuce scientific name you studied. "

Abstract Line 30. Write the scientific name of the lettuce  variety/cultivars you studied.

Pay attention to the use of points, commas, spaces, and capital letters in the whole manuscript.

Before the use of PGPR, you must explain it in the text

Introduction. Explain the problem in more detail. Provide references and explain what makes your study different.

Materials and methods. In each method explain with a phrase the reason why you performed this test.

Results: Phosphorus solubilization test should be suppourte by fotographs of halozone formation. Please share it as regular or supplementary figure.

In IAA production, After 30 minutes of incubation at 708 room temperature in the dark, IAA production was detected spectrophotometrically by 709 measuring the absorbance at 530 nm. But acutally here Pink colour is the indicator. Please share the pink colour in test tubes as supplementary figure

Conclusions must be clear and must show the novelty of your article.

Author Response

The authors declaimed that Our results show that specific Bacillus species from the Atacama Desert enhance drought stress tolerance in lettuce plants by promoting several beneficial plant traits that facilitate water absorption, and nutrient uptake, which support its use as potent bioinoculants.

The authors need to revise the English grammar and spelling of the article. There are many incomprehensible phrases.

R: Done, thank you for your suggestion. We carry out a comprehensive review of the English grammar.

I do not understand the novelty of the article.

R: Thank you for the comment, done. This is explained in the introduction; please see lines 102-118.

Title. What do you mean? You must specify the lettuce scientific name you studied.

R: Thank you for the comment, done. Please see the abstract and material and methods.

Abstract Line 30. Write the scientific name of the lettuce variety/cultivars you studied.

R: Done.

Pay attention to the use of points, commas, spaces, and capital letters in the whole manuscript.

R: Thanks for this valuable observation, done.

Before the use of PGPR, you must explain it in the text.

R: Done, se line 65.

Introduction. Explain the problem in more detail. Provide references and explain what makes your study different.

R: Done; please note that this information is included in the introduction.

Materials and methods. In each method explain with a phrase the reason why you performed this test.

R: Done.

Results: Phosphorus solubilization test should be suppourte by fotographs of halozone formation. Please share it as regular or supplementary figure.

R: Done, please see Supplementary Material, also we included N2 fixation and ACC deaminase activity.

In IAA production, after 30 minutes of incubation at room temperature in the dark, IAA production was detected spectrophotometrically by measuring the absorbance at 530 nm. But acutally here Pink colour is the indicator. Please share the pink color in test tubes as supplementary figure.

R: We apologize because we cannot fulfill the request as we do not have a photographic record.

Conclusions must be clear and must show the novelty of your article.

R: Thanks for this observation, please see the introduction and material and methods.

Reviewer 3 Report

Comments and Suggestions for Authors

There are many English language mistakes and some semanthic errors. However, this problem has been reported to the Editor.

In the Abstract the sentence "Seven strains of Bacillus spp. were isolated from the rhizosphere of the Chilean endemic plants 28 Metharme lanata and Nolana jaffuelii, using the 16s rRNA gene"  should be modified into "Seven strains of Bacillus spp. were isolated from the rhizosphere of the Chilean endemic plants 28 Metharme lanata and Nolana jaffuelii, and then identified using the 16s rRNA gene".

In the Introduction, and maybe throughout the manuscript, the genus Bacillus is followed by spp. which should be in italics.

In the Discussion many sentences contain language mistakes.

For example, Lines 461-465: Among this, the strain ATMLC92021 and ATMLC152021  was closely related with Bacillus subtilis which some strains of these species has been reported improve the drought tolerance in sugarcane, fenugreek and wheat. In the  case of strain ATMLC102021 were closely related to Bacillus tequilensis clade, this species  described as osmotic stress tolerant .

Lines 476-478:For strain ATNJC12015, ATNJC22015, ATMLC42021 was grouped in Bacillus 476 atrophaeus, Bacillus ginsengihumi and Bacillus frigoritolerans respectively. For these, species 477 there was not previously works to report an improvement of drought tolerance in plants.

Comments on the Quality of English Language

There are many English language mistakes and some semanthic errors.

Author Response

There are many English language mistakes and some semanthic errors. However, this problem has been reported to the Editor.

R: Thanks for this valuable observation; it has been improved throughout the document.

In the Abstract the sentence "Seven strains of Bacillus spp. were isolated from the rhizosphere of the Chilean endemic plants 28 Metharme lanata and Nolana jaffuelii, using the 16s rRNA gene"  should be modified into "Seven strains of Bacillus spp. were isolated from the rhizosphere of the Chilean endemic plants 28 Metharme lanata and Nolana jaffuelii, and then identified using the 16s rRNA gene".

R: Done.

In the Introduction, and maybe throughout the manuscript, the genus Bacillus is followed by spp. which should be in italics.

R: Done.

In the Discussion many sentences contain language mistakes.

R: Thank you for this valuable observation, we carry out a complete review of the English grammar.

For example, Lines 461-465: Among this, the strain ATMLC92021 and ATMLC152021 was closely related with Bacillus subtilis which some strains of these species have been reported improve the drought tolerance in sugarcane, fenugreek and wheat. In the case of strain ATMLC102021 were closely related to Bacillus tequilensis clade, this species described as osmotic stress tolerant.

R: Done.

Lines 476-478: For strain ATNJC12015, ATNJC22015, ATMLC42021 was grouped in Bacillus 476 atrophaeus, Bacillus ginsengihumi and Bacillus frigoritolerans respectively. For these, species 477 there was not previously works to report an improvement of drought tolerance in plants.

R: Done.
